# Rapid LC-MS/MS Bosutinib Quantification with Applications in Metabolic Stability Estimation

**DOI:** 10.3390/molecules28041641

**Published:** 2023-02-08

**Authors:** Mohamed W. Attwa, Mohammed M. Alanazi

**Affiliations:** Department of Pharmaceutical Chemistry, College of Pharmacy, King Saud University, Riyadh 11451, Saudi Arabia

**Keywords:** bosutinib, rapid LC-MS/MS, validation, metabolic stability, in vitro half-life, intrinsic clearance

## Abstract

Bosutinib (BOS) is FDA approved drug for the treatment of chronic phase (CP) Philadelphia chromosome-positive (Ph+) chronic myelogenous leukemia (CML). We report a fast, sensitive, and simple LC-MS/MS method, validated for the determination of BOS in human liver microsomes, utilizing tofacitinib (TOF) as the internal standard. The separation of BOS and TOF was done using a 1.8 μm C18 column (2.1 × 50 mm) at room temperature using the isocratic elution system of acetonitrile–water (30:70, *v*/*v*) containing 0.1 M formic acid at a flow rate of 0.15 mL/min, and a triple-quadrupole tandem mass spectrometer (TQD-MS) with an electrospray ionization (ESI) source that was operated in the positive ion mode. The method was validated according to the European Medicines Agency, and the rapid and specific quantification of BOS in human liver microsomes was achieved in the range of 5–200 ng/mL, with a determination coefficient of 0.999. Intra- and inter-day accuracy and precision values were <4% in all cases. The procedure is rapid, specific, reliable, and can be applied in metabolic stability evaluations since it is the first LC-MS/MS method specific to BOS quantification. The metabolic stability assessment of BOS showed high CL_int_ (34.3 µL/min/mg) and short in vitro t_1/2_ values of 20.21 min, indicating that BOS may be rapidly eliminated from the blood by the liver.

## 1. Introduction

Chronic myeloid leukemia (CML) is the outcome of constitutive tyrosine kinase BCR-Abl enzyme activity, the product of the bcr-abl gene fusion present in the Philadelphia chromosomes of patients whom suffer from CML [1]. Imatinib is considered a selective inhibitor of BCR-Abl and its introduction to patients represented an outstanding improvement in CML therapy [2]. The platelet-derived growth factor receptor (PDGFR) and the tyrosine kinase c-Kit are strongly inhibited by imatinib, which is currently utilized to treat malignancies produced by the dysregulated forms of these proteins [3,4]. Despite the success of imatinib in CML treatment, some patients experience clinical relapse as they eventually establish resistance to imatinib treatment [5]. The occurrence of imatinib resistance has led to research on additional inhibitors of BCR-Abl, and the second-generation inhibitors (nilotinib and dasatinib) were recently approved for use in imatinib-resistant CML patients, in addition to front-line therapy [6,7]. Though nilotinib and dsatinib and are active against most imatinib-resistant BCR-Abl mutations, neither drug is effective against BCR-Abl having the common T315I mutation. Patients who primarily respond to dasatinib therapy and consequently relapse have been shown to carry new BCR-Abl mutations, revealing the emergence of clinical resistance to second-generation inhibitors [8]. Therefore, the research on additional BCR-Abl inhibitors is important, both to fight resistance and to expand the therapeutic choices of patients with CML.

The Food and Drug Administration (19/12/2017) granted accelerated approval for bosutinib (BOSULIF), which was developed by Pfizer Inc. for the treatment of patients with newly diagnosed chronic phase (CP) Philadelphia chromosome-positive (Ph+) CML [9]. Bosutinib (BOS; Figure 1) is a second-generation dual Abl/Src inhibitor that shows potent inhibition of the growth of CML cells in vitro, is also active against multiple imatinib-resistant BCR-Abl mutations, and has proven efficacy in current clinical trials for imatinib-resistant CML [10,11,12]. The most common side effect of BOS is diarrhea and it can be avoided by concurrent antidiarrheal medication. Other minor side effects of BOS include possible dermatological problems, grade 2 diarrhea, grade 1 fatigue associated with secondary dehydration caused by diarrhea, grade 1 skin rash, grade 1 AST elevation, and grade 2 vomiting, which indicate the safety of BOS if compared to imatinib [13,14].

The metabolic stability of a chemical compound or drug is defined as its susceptibility to metabolism and is expressed as the in vitro half-life [t_1/2_] and the intrinsic clearance [CL_int_]. The half-life [t_1/2_] is the time required for 50% removal of the parent drug. Intrinsic clearance [CL_int_] is the ability of the liver to metabolize or eliminate drugs in the blood. The two parameters are computed following the “in vitro half-life” approach based on the “well-stirred” model [15,16]. As the “well-stirred” model is the most frequently used approach in in vitro drug metabolism prediction, the derived in-vitro-calculated parameters can be used for the prediction of various in vivo physiological parameters, including potential toxicity and accumulation [17,18].

The study of the metabolic stability of BOS is important during drug discovery for the development of drugs with better metabolic stability profiles [19]. Rapidly metabolized drugs exhibit a decrease in in vivo bioavailability, leading to the shorter duration of their action [20]. No analytical method has been developed for quantifying BOS in human liver microsomes (HLMs) or for the metabolic stability estimation of BOS. Accordingly, this study focuses on the quantification of BOS in spiked HLMs using tofacitinib as the internal standard (TOF; IS) over a very short run time (5 min), which permits its application in metabolic stability estimations. This study aimed to develop and validate a reliable LC-MS/MS method. Protein precipitation using acetonitrile (ACN) was used for analytes’ (BOS and TOF) extraction from the HLM matrix. All analytical parameters, such as calibration, recovery, accuracy, and precision, were determined according to the FDA guidelines. Finally, a metabolic stability experiment with BOS in HLM was performed successfully using the established LC-MS/MS method guided by the in silico assessment of its stability.

## 2. Results

### 2.1. In Silico BOS Metabolic Stability

The BOS (C_26_H_29_Cl_2_N_5_O_3_) metabolic landscape shows the degree of the metabolic instability of the active sites of BOS, metabolized by the CYP enzymes [21,22,23]. The sites were classified as having the highest degree of metabolic instability (labile; orange color) to the lowest degree of metabolic stability (mod. labile; yellow color) or being metabolically stable (stable; black color). Figure 2 shows the metabolic landscape of BOS, where C1, C7, C3, C4, and C6 of the methyl piperazine group are labile to metabolism, while the C36 of the methoxy group attached to the dichlorophenyl group and the C36 of the methoxy group attached to the quinoline–carbonitrile group are moderately labile. These findings (CSL:0.9966) reveal the high metabolic instability of BOS; therefore, the established methodology was applied to assess the metabolic stability of BOS.

The metabolic instability of BOS may be attributed to the methyl piperazine group, as proposed by the StarDrop software (P450 metabolism model). These results indicate the importance of performing an in vitro metabolic stability assessment of BOS and the need to establish an LC-MS/MS method for the quantification of BOS in HLM matrices.

### 2.2. LC-MS/MS Method Development

TOF was selected as the IS in BOS quantification in the current LC-MS/MS method, because the protein precipitation extraction methodology using ACN can be used for the extraction of both analytes (BOS and TOF) from the HLM matrix. The extraction recoveries of BOS and TOF were 102.7 ± 3.12% and 101.63 ± 2.58%, respectively. The elution times of TOF and BOS were 1.7 and 3.3 min, respectively, revealing reasonable separation. BOS and TOF are anti-cancer drugs that are not prescribed together; consequently, the established LC-MS/MS method could be used for pharmacokinetic or therapeutic drug monitoring (TDM) studies of BOS.

Under the chromatographic conditions described in the experimental part, BOS and TOF were well resolved, and the carryover effect was minimal in both the negative control (HLM matrix) and positive control (HLM matrix plus TOF). Figure 3 shows the overlaid MRM chromatograms of the BOS calibration levels in addition to TOF (IS) in the HLM matrix.

The BOS and TOF mass spectra exhibited a parent ion peak at *m*/*z* 530 and 313.2, respectively. The MS/MS fragmentation study involved the isolation of *m*/*z* 530 (BOS) and 313.2 (TOF) in the first quadrupole mass analyzer (Q1), followed by fragmentation (q2) in the collision cell using high-purity nitrogen as a collision gas. The scanning of the highest-intensity and most reproducible fragments in the second quadrupole mass analyzer (Q3) revealed *m*/*z* 113 and 141 for BOS, and *m*/*z* 165 and 148.9 for TOF, which were selected as daughter ion peaks in the MRM detection mode, as shown in Figure 4.

### 2.3. Method Validation

#### 2.3.1. Specificity

The developed LC-MS/MS method showed high specificity for BOS quantification, as no interference was observed from the constituents of the HLM matrix at the elution times of BOS and TOF. The mass detector did not exhibit any carryover effects from the samples. TOF and BOS were well separated using the optimized chromatographic parameters, with elution times of 1.7 and 3.2 min, respectively.

#### 2.3.2. Linearity and Sensitivity

The statistical analysis of BOS quantification data was performed using the least-squares method. The results of the six BOS calibration curves showed linearity in the range of 5–200 ng mL^−1^ for BOS, with a determination coefficient (r^2^) ≥ 0.9991. The mean calibration curve of the BOS standard solutions in the HLM matrix was described by y = 2.5773x + 1.31. The standard deviation (SD) values of each conc. level (six replicates) did not exceed 1.08%. The limit of detection (*LOD*) and limit of quantitation (*LOQ*) were calculated using Equation (1).
(1)LOQ OR LOD=k υM 
where k is equal to 10 and 3.3 for the *LOQ* and *LOD*, respectively; υ is the SD of the intercept, and M is the slope of the regression line of the established calibration curve. The *LOQ* and the *LOD* were 0.97 and 0.32 ng/mL in the HLM matrix, respectively. Calibration standards and quality control levels of BOS in the HLM matrix (ten points) were back-calculated to ensure the best performance of the supposed method. The accuracy and precision for BOS in spiked HLM matrix samples were 97.65–101.55% and 0.27–1.53%, respectively (Table 1).

#### 2.3.3. Precision and Accuracy

The accuracy and precision of the presented chromatographic procedure was confirmed using the intra-day and inter-day accuracy and precision outcomes of the BOS QC samples. The percentages of the relative error (% RE) and relative standard deviation (% RSD) were utilized to evaluate the analytical method accuracy and precision, respectively, using Equation (2). The results for precision and accuracy were within the permitted range according to the EMA guidelines [10], as displayed in Table 2.
(2)$$\%\;Error=\left[\frac{\left(Mean\;measured\;concentration-nominal\;concnetration\right)}{nominal\;concentration}\right]\times 100$$

### 2.4. In Vitro Metabolic Stability of BOS

HLM (1 mg protein) was used in the HLM matrix so as to avoid nonspecific protein binding. BOS (1 µM) was utilized in incubation in the HLM matrix (so as to be lower than the Michaelis–Menten constant). The BOS conc. was computed using the linear curve regression equation of a freshly constructed calibration curve. The BOS metabolic stability curve was established by plotting the remaining BOS percentage (y-axis) against the incubation time in minutes (x-axis) (Figure 5A). The linear part of the metabolic stability curve (0–70 min) was used to establish another curve of the natural logarithm (Ln) of the remaining BOS against the selected incubation [n time range (0–20 min) (Figure 5B). The regression equation of the Ln curve was y = −0.034339x + 4.592, with R² = 0.9964. The slope of the linear part of the Ln curve (0.0343) represents the rate constant for BOS metabolism, which was utilized to compute the in vitro t_1/2_ (20.21 min) of BOS using Equation (3) (Table 3). The BOS intrinsic clearance (CL_int_) was computed after the in vitro t_1/2_ method using Equation (4). The CL_int_ of PMB was 34.3 µL/min/mg. Based on these results, it can be proposed that BOS is characterized by a moderate rate of extraction from the body and it is proposed to moderately accumulate in the body with reasonable bioavailability, compared with other tyrosine kinase inhibitors (e.g., dacomitinib). By using simulation software and the Cloe PK, these outcomes could also be used to predict the in vivo pharmacokinetics of BOS [24].
(3)In vitro t ½=ln2Slope

CL_int_ (µL/min/mg) was calculated using Equation (5):(4)$${\mathrm{CL}}_{\mathrm{int},\;}=\frac{0.693}{In\;vitro{\;t\;}_{½}}\;.\frac{µL\;\mathrm{incubation}}{\mathrm{mg}\;\mathrm{microsomes}}\;$$

## 3. Materials and Methods

### 3.1. Chemicals and Reagents

The pooled HLM matrix from the human liver (Product Number: M0567) was purchased from Sigma-Aldrich (St. Louis, MO, USA). HLM was stored at −70 °C until use. The reported protein content of the HLM pool (20 mg/mL) was shipped in 250 mM sucrose to keeps HLMs active. BOS (B-1788; purity: 99.89%) and TOF (T-1377; IS; purity: 99.84%) reference powders were obtained from LC Laboratories (Woburn, MA, USA), Purified water was obtained using a Milli-Q plus purification system that was procured from Millipore (Billerica, MA, USA). HPLC-grade acetonitrile (ACN) and formic acid were obtained from Sigma-Aldrich and VWR International (West Chester, PA, USA).

### 3.2. In Silico BOS Metabolic Stability Evaluation

The in silico stability of BOS towards metabolism was determined utilizing the P450 metabolism module of the StarDrop software package from Optibrium Ltd. (Cambridge, MA, USA). To determine the metabolic lability of BOS, the site labilities of individual atoms can be combined to calculate the composite site lability (CSL), which indicates the overall metabolic stability of BOS, as explained by Equation (5):(5)CSL =ktotalktotal+kw
where *kw* is the rate constant for water formation.

The BOS (CN1CCN(CC1)CCCOc2cc3c(cc2OC)c(c(cn3)C#N)Nc4cc(c(cc4Cl)Cl)OC) SMILES format was uploaded to the StarDrop P450 metabolism module for CSL prediction. CSL is considered a crucial parameter in predicting the metabolic rate of BOS before establishing in vitro experiments to validate the significance of the current work. The CSL values in the metabolic landscape were used as indicators of BOS’ metabolic stability.

### 3.3. Instrumentation and Conditions

The analysis of BOS (C_26_H_29_Cl_2_N_5_O_3_) and TOF (C_16_H_20_N_6_O) was performed in the positive mode (ESI+). The mass spectrometry analyses were performed using a triple-quadrupole (TQD) mass analyzer (MS/MS), and the spectrometric parameters were adjusted to detect and analyze BOS and TOF (IS) with good accuracy and sensitivity. Tuning was performed utilizing the IntelliStart® module of the QuanLynx software, which was optimized manually in the infusion mode of fluidics to increase the peak selectivity and intensity of BOS and TOF. Argon (0.14 mL/min) was used as the collision gas in the quadrupole 2 (collision cell) for dissociation of the parent ion peak to fragment ions. High-purity nitrogen gas (650 L/h) was used as the drying gas at 350 °C. The cone gas flow rate was maintained at 100 L/h. The MRM mode was utilized for quantification to increase the selectivity and sensitivity of the developed LC-MS/MS method.

The MRM mass transitions for BOS (Rt: 3.3 min) were 530 → 141 (CV: 45 V and CE: 25 V) and 530 → 113 (CV: 45 V and CE: 20 V) (Figure 4A). The TOF peak (Rt: 1.7 min) was estimated using the selected MRM mass transitions: 313 → 165 (CV: 35 V and CE: 20 V) and 313 → 149 (CV: 40 V and CE: 24 V) (Figure 4B). The MRM mode was used for the detection of BOS and TOF to eliminate interference from the HLM matrix, which elevated the sensitivity of the established LC-MS/MS method.

LC analytical chromatographic parameters including the resolution of the target analytes (BOS and TOF), such as the mobile phase composition, stationary phase nature, and pH, were optimized. The mobile phase consisted of 30% ACN and 70% aqueous solution at a flow rate of 0.15 mL/min. Increasing ACN % generated overlapped peaks and a poor resolution, while decreasing ACN% generated long elution times. The pH of the 0.1% formic acid in water (aqueous solution) was 3.2, as an increased pH value caused a long elution time and chromatographic peak tailing. Different stationary phases were examined, such as polar columns (HILIC columns). However, neither BOS nor TOF was retained on the chromatographic column, and the best results were achieved through the use of a C18 column (2.1 × 50 mm, 1.8 μm) (Agilent, Santa Clara, CA, USA) at 22 ± 2 °C. Injection volume and run time were 5 min and 5 µL, respectively.

### 3.4. Preparation of the Standard Solutions

BOS and TOF showed reasonable DMSO solubility at ≥46 mg/mL (86.72 mM) and 125 mg/mL (400.17 mM; under ultrasonication), respectively. The BOS stock standard solution (1.0 mg mL^−1^) was generated in DMSO and was further diluted to create a working standard solution (1 µg mL^−1^). To obtain the TOF (IS) stock solution, the reference TOF powder was dissolved in DMSO to obtain a conc. of 0.1 mg/mL. Then, 100 μL of this stock solution was diluted using the mobile phase to 10 mL to generate a working solution of 1 µg/mL.

### 3.5. Sample Preparation and Construction of the Calibration Curve

DMSO is able to stop metabolic reactions even at a 0.2% concentration. DMSO (2%) was used during the validation steps as a quenching agent for the HLM matrix under slight heating at 50 °C for 5 min because of the solubility of BOS and TOF. The HLM matrix was prepared by diluting 30 µL deactivated HLMs (1 mg protein/1 mL) to 1 mL of metabolic medium (0.1 M sodium phosphate buffer (pH 7.4) and 1 mM NADPH). Ten calibration points, 5, 10, 15 (low-quality control; LQC), 20, 30, 40 (medium-quality control; MQC), 50, 100, 150 (high-quality control; HQC), and 200 ng/mL were utilized to generate the calibration curve including the three selected quality controls. Then, 100 μL of TOF (1 µg/mL) was added to each level. A protein precipitation extraction methodology was used to extract BOS and TOF from the HLM matrix. Then, 2 mL of acetonitrile was added to 1 mL of each sample, vortexed for 1 min, centrifuged at 14,000 rpm (12 min at 4 °C), and the supernatant was filtered using a 0.22 µm syringe filter. The filtrate was placed in 1.5 mL HPLC vials and 5 µL was injected into the LC-MS/MS system. The quality control (QC) samples were prepared using the same procedure. A calibration curve was constructed by plotting the conc (x-axis) against the peak area ratio of BOS to TOF (y-axis). The linearity of the method was assessed using the characteristic linear regression parameters.

### 3.6. BOS Metabolic Stability

The metabolic stability profile of BOS, including the two factors (CL_int_ and in vitro t_1/2_), was defined by the quantification of the percentage remaining of the BOS conc. after incubation with the HLM matrix that was composed of 30 µL of HLM, NADPH (cofactor for metabolic reaction), and 3.3 mM MgCl_2_ in a 0.1 M sodium phosphate buffer (pH 7.4) for 70 min. In the first step, pre-incubation of BOS 1 µM was performed using the HLM matrix (without NADPH) for 10 min, to adjust the optimal conditions for the initiation of the metabolic reactions. In the second step, NADPH (1 mM) was added to the HLM mixture to initiate the metabolic reaction. To verify the outcomes, the previous metabolic experiments were repeated three times. TOF WK3 (100 µL, 1 µg/mL) was added to the HLM mixture as the IS shortly prior to the stopping of the metabolic reaction, to avoid the effect of metabolic enzymes on the IS conc. In the third step, termination of the ongoing metabolic incubation was performed at certain time points (0, 2.5, 7.5, 15, 20, 30, 40, 50, 60, and 70 min) through the addition of 2 mL of ice-cold ACN. The extraction steps were performed as described in Section 3.5. Data analysis was done using the QuanLynx module included in the MassLynx 4.1 software package. The conc. of the BOS at specific time points was computed, and the BOS metabolic stability curve was constructed. It was supposed that the conc. of BOS at 0 min was 100% and the remaining BOS% was plotted against time. From this curve, linear range points were chosen to construct a metabolic curve exhibiting the natural logarithm of the percentage of the remaining BOS over time. The slope of the linear part of the curve indicated the rate constant of BOS metabolism and was utilized to compute the in vitro t_1/2_ using Equation (3). CL_int_ (µL/min/mg) was computed using Equation (4). CL_int_ was then scaled to in vivo clearance using the HLM protein concentration per gram of liver and the average liver weight reported in the literature.

## 4. Conclusions

A rapid LC-MS/MS method was developed and validated for the quantification of BOS in HLM matrices, with good linearity in the range of 5–200 ng mL^−1^. The sample preparation procedure is simple, with a short run time of 5 min. The proposed procedure involves eco-friendly elution with reduced consumption of organic solvents along with low running costs. Our validated LC-MS/MS method was applied in evaluating the metabolic stability of BOS, revealing that BOS exhibited moderate CL_int_ (34.3 µL/min/mg) and in vitro t_1/2_ values of 20.21 min, suggesting a high hepatic clearance rate. Consequently, acceptable in vivo bioavailability can be predicted. Based on these results, we propose that BOS can be administered to patients without dose accumulation in human blood. The in vitro metabolic experiment results matched the in silico predictions; therefore, they could be used to predict the metabolic stability of other drugs. Future studies are required to verify the in silico prediction method for in vivo therapeutic drug monitoring.

## Figures and Tables

**Figure 1 molecules-28-01641-f001:**
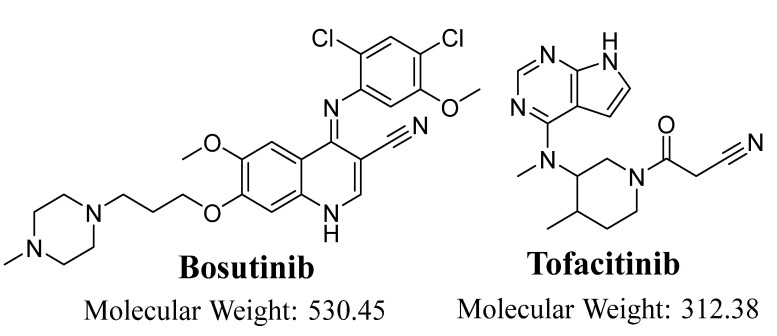
Chemical structures of bosutinib (BOS) and tofacitinib (TOF; IS).

**Figure 2 molecules-28-01641-f002:**
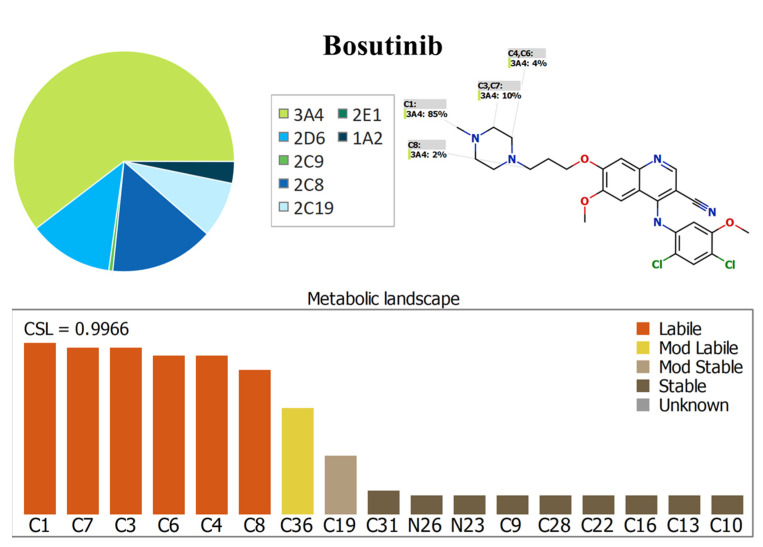
The predicted metabolic stability landscape of bosutinib (BOS) utilizing the P450 metabolism module (StarDrop software package).

**Figure 3 molecules-28-01641-f003:**
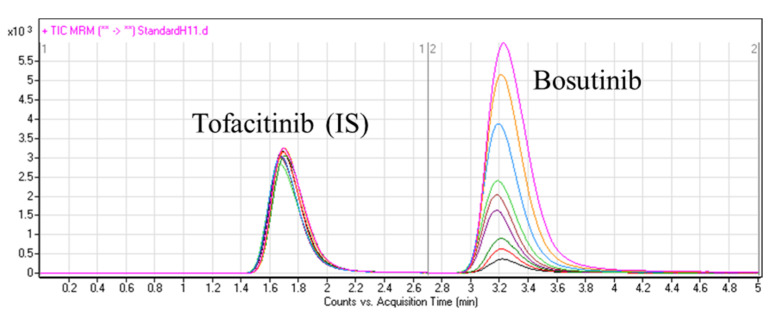
MRM chromatograms of BOS calibration standards and TOF (IS) in spiked HLM matrix samples.

**Figure 4 molecules-28-01641-f004:**
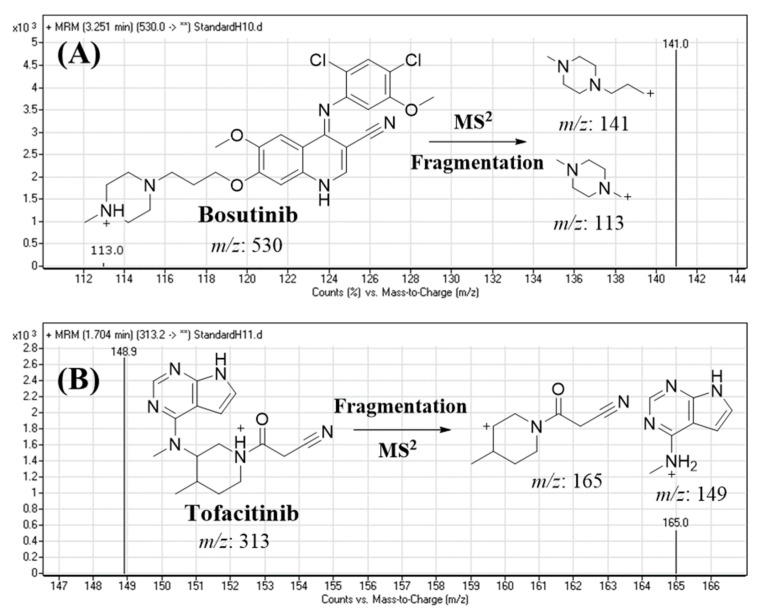
MRM mass spectra of bosutinib (**A**) and tofacitinib (**B**).

**Figure 5 molecules-28-01641-f005:**
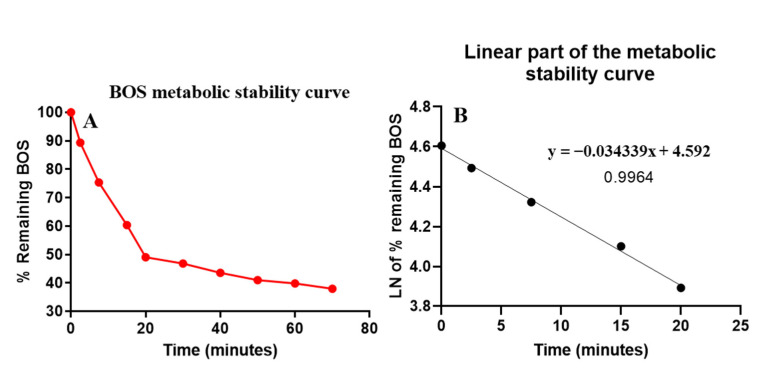
(**A**) BOS metabolic stability curve in HLM and (**B**) the linear part of the metabolic stability curve of BOS.

**Table 1 molecules-28-01641-t001:** Data of calibration standards and quality control levels of BOS in the HLM matrix.

Nominal Conc. (ng/mL)	Mean ^a^	Standard Deviation (SD)	Precision (%)	Accuracy (%)
5	5.07	0.05	1.00	101.43
10	10.16	0.14	1.35	101.55
15	15.11	0.04	0.27	100.74
20	19.53	0.30	1.53	97.65
30	29.70	0.19	0.65	99.01
40	39.40	0.20	0.52	98.51
50	50.56	0.20	0.40	101.11
100	100.70	0.34	0.33	100.70
150	150.04	1.08	0.72	100.03
200	199.66	0.76	0.38	99.83

^a^ Average of six determinations.

**Table 2 molecules-28-01641-t002:** Inter-day and Intra-day accuracy and precision of BOS (QC) samples.

Day of Analysis	Measured Concentration of Bosutinib in HLM Matrices					
	LQC (15 ng mL^−1^)		MQC (50 ng mL^−1^)		HQC (150 ng mL^−1^)	
Day 1	15.18		50.61		151.22	
	15.08		49.96		149.09	
	15.14		50.77		150.52	
	15.08		50.74		149.49	
	15.10		50.62		148.16	
	15.14		50.32		148.86	
	15.00		50.60		151.22	
	15.04		50.31		148.94	
	15.05		49.90		148.39	
	15.12		50.63		151.20	
	15.12		50.30		148.84	
	15.15		50.79		150.61	
Day 2	15.00		50.22		150.25	
	15.03		50.00		150.46	
	15.07		50.01		150.12	
	15.04		50.05		150.13	
	15.02		50.08		150.31	
	15.16		50.31		149.90	
Day 3	15.12		50.18		150.48	
	15.05		50.05		150.12	
	15.07		50.15		149.79	
	15.00		49.99		150.12	
	15.01		50.15		150.43	
	15.15		50.04		150.51	
	Intra-day *	Inter-day **	Intra-day	Inter-day	Intra-day	Inter-day
Mean	15.10	15.08	50.46	50.28	149.71	149.97
SD	0.05	0.06	0.30	0.29	1.16	0.86
%RSD	0.33	0.37	0.60	0.57	0.78	0.57
%RE	0.66	0.53	0.92	0.56	−0.19	−0.02

* Average of 12 determinations in the same day. ** Average of six determinations in three following days.

**Table 3 molecules-28-01641-t003:** Parameters of the BOS metabolic stability curve.

Time (min.)	Mean ^a^ (ng/mL)	X ^b^	LN X ^c^	Analytical Parameters
**0**	195.00	100	**4.61**	Linear regression equation: y = −0.034339x + 4.592
**2.5**	174.27	89.36	**4.49**	
**7.5**	146.89	75.32	**4.32**	R² = 0.9964
**15**	117.69	60.35	**4.10**	
**20**	95.60	49.02	**3.89**	Slope: −0.0254
30	91.37	46.85	3.85	
40	85.00	43.58	3.77	t_1/2_: 20.21 min
50	79.93	40.98	3.71	CL_int_: 34.3 µL/min/mg
60	77.72	39.85	3.69	
70	74.07	37.98	3.64	

^a^ Mean of three repeats. ^b^ X: Mean of the BOS remaining percentage from three repeats. ^c^ The linear range is exhibited with bold font.

## Data Availability

All data are available within the manuscript.

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
