# Peer review of "Rapid LC-MS/MS Bosutinib Quantification with Applications in Metabolic Stability Estimation"

_molecules, 2023, doi:10.3390/molecules28041641_

Round 1

Reviewer 1 Report

The manuscript represents the study of the metabolic stability of bosutinib using a validated LC-MS/MS method. The work was very good and scientifically sound and the experiments were performed in a perfect way and can be repeated.

There are some minor concerns that have to fixed.

1- The author should revise the whole manuscript for some minor mistakes in spelling and language. 

2- In section 2.6 should be corrected as there is no AVB or GSB in the current study. I think it should be replaced by BOS and TOF. This paragraph should be corrected and rephrased.

3- In figure 3, the structure should be inverted by 90 degrees.

4- In figure 2, MS2 should be replaced by MRM. 

5- The first sentence of the abstract should be about the drug definition and importance.

6- The catalog number of reference powder should be added.

7- The way of writing mLmin-1 should be mL/min

8- All journal names should be abbreviated

3- 

Author Response

Authors’ response

We thank the editor this opportunity to improve our manuscript and be considered again for publication in Molecules Journal. We give below detailed answers to each question raised by reviewer # 1. We mark all reply to the comments by red color in the revised manuscript.

The manuscript has been edited to ensure language and grammar accuracy by Editage (a brand of Cactus Communication). We attached a certificate of English editing.

 Reviewer # 1

Comments and Suggestions for Authors

The manuscript represents the study of the metabolic stability of bosutinib using a validated LC-MS/MS method. The work was very good and scientifically sound and the experiments were performed in a perfect way and can be repeated.

There are some minor concerns that have to fixed.

Authors’ response

We appreciate the reviewer’s words and his/her suggestions to improve our manuscript. We give below our answer to his/her concerns.

Point # 1  

1- The author should revise the whole manuscript for some minor mistakes in spelling and language. 

Authors’ response

The manuscript has been edited to ensure language and grammar accuracy by Editage (a brand of Cactus Communication). We attached a certificate of English editing.

2- In section 2.6 should be corrected as there is no AVB or GSB in the current study. I think it should be replaced by BOS and TOF. This paragraph should be corrected and rephrased.

Authors’ response

We appreciate the reviewer’s comment and his/her suggestions to improve our manuscript.

We rearranged the manuscript according to journal guidelines, so section 2.6 becomes 4.6 as journal outlines.

Sorry for this typo mistakes and we rearranged the whole paragraph.

We updated the following paragraph in the revised manuscript:

“3.6. BOS metabolic stability

The metabolic stability profile of BOS, including the two factors (CLint and in-vitro t1/2), was defined by the quantification of the percent remaining of BOS conc. after incubation with HLMs matrix that composed of 30 µL of HLM, NADPH (cofactor for metabolic reaction), and 3.3 mM MgCl2 in a 0.1 M sodium phosphate buffer (pH 7.4) for 70 min. The first step, pre-incubation of BOS 1 µM was performed using HLMs matrix (without NADPH) for 10 min, to adjust the optimal conditions for initiation of the metabolic reactions. The second step, NADPH (1 mM) was added to the HLMs mixture for initiating the metabolic reaction. To verify the outcomes, the previous metabolic experiments were repeated three times [34]. TOF WK3 (100 µL, 1 µg/mL) was added to the HLMs mixture as the IS just prior to the stopping of the metabolic reaction, to avoid the effect of metabolic enzymes on the IS conc. The third step, termination of the ongoing metabolic incubation was done at certain time points (0, 2.5, 7.5, 15, 20, 30, 40, 50, 60, and 70 min) through the addition of 2 mL of ice-cold ACN. The extraction steps were performed as described in section No 3.5. Data analysis was done using the QuanLynx module involved in the MassLynx 4.1 Software package. The conc. of the BOS at specific time points was computed, and the BOS metabolic stability curve was constructed. It was supposed that the conc. of BOS at 0 min was 100% and the remaining BOS% was plotted against time. From this curve, linear range points were chosen to construct a metabolic curve exhibiting the natural logarithm of the percentage of the remaining BOS over time. The slope of the linear part of the curve indicates the rate constant of BOS metabolism and was utilized to compute the in-vitro t1/2 using equation (3) [35]. CLint (µL/min/mg) was computed using equation (4). CLint was then scaled to in-vivo clearance using the HLM protein concentration per gram of liver and the average liver weight reported in the literature [36].

3- In figure 3, the structure should be inverted by 90 degrees.

Authors’ response

Sorry figure 3 could not be changed as this is generated by the software also the marking above the image will not be inverted.

4- In figure 2, MS2 should be replaced by MRM. 

Authors’ response

We rearranged the manuscript according to journal guidelines.

So figure 2 becomes figure 4.

We updated the following sentence in the revised manuscript as requested:.

“Figure 4. MRM mass spectra of Bosutinib (A) and Tofacitinib (B)”

5- The first sentence of the abstract should be about the drug definition and importance.

Authors’ response

We updated the following sentence in the revised manuscript as requested:.

“Bosutinib (BOS) is an approved drug for the treatment of chronic phase (CP) Philadelphia chromosome-positive (Ph+) chronic myelogenous leukemia (CML)”

6- The catalog number of reference powder should be added.

Authors’ response

We added the required informations and we updated the following paragraph in the revised manuscript as requested.

“The pooled HLM matrix from human livers (Product Number: M0567) was pur-chased from Sigma-Aldrich (St. Louis, MO, USA) and stored at −70 °C. The reported pro-tein content of the HLM pool was 20 mg/mL in 250 mM sucrose. BOS (B-1788); purity: 99.89% and TOF (T-1377); IS; purity: 99.84% reference powders were obtained from LC Laboratories (Woburn, MA, USA), HPLC-grade acetonitrile (ACN) and formic acid were obtained from Sigma-Aldrich and VWR International (West Chester, PA, USA). Purified water was obtained using a Milli-Q plus purification system (Millipore, Waters, Billerica, MA, USA).”

7- The way of writing mLmin-1 should be mL/min

Authors’ response

We changed the mLmin-1 to mL/min as requested and we checked and updated the revised manuscript as requested..

8- All journal names should be abbreviated

Authors’ response

We abbreviated journal names as requested and we checked and updated the revised manuscript as requested..

Kind regards

Dr. Mohamed Attwa

Reviewer 2 Report

This research article is aiming at developing a rapid LC-MS/MS method to characterize bosutinib after investigating calibration range, intra- and inter-day accuracy and precision and other analytical standards. Although the method design is straightforward and correct, this work lacks novelty, experimental details and relevance to this journal. In addition, this manuscript has various flaws in its English style and is not qualified as an acceptable form, especially some sentences are really redundant and confusing. Please see some detailed suggestions below.

Comments:

1. P.1 title, “rapid validated” has a grammar mistake. This title is confusing with the combination of these three parts. Please rephrase.

2. P.1 short title, should be “Bosutinib’s metabolic stability” or “The metabolic stability of Bosutinib”.

3. P.2, what’s meaning of putting its full name in the beginning of the paragraph while lacking any elaboration? “Bosutinib (BOS; bosutinib hydrate, 4‐[(2, 4‐Dichloro‐5‐methoxyphenyl) amino]‐6‐methoxy‐7‐[3‐(4‐methylpiperazin‐1‐yl) propyloxy] quinoline‐3‐carbonitrile monohydrate).” 

4. P.2, the main text should also contain “Fig.1” which can give readers its reference.

5. P.2, delete one “in” in “used approach in in vitro”.

6. P.2, “the derived parameters can be used for extrapolation to various in vivo physiological parameters” sounds cryptic, please improve wording.

7. P.2, please rephrase and rearrange this paragraph in the main text, “Diarrhea is the most common side effect in patients taking bosutinib and concurrent antidiarrheal medication may reduce this side effect”. I understand that authors intend to show the importance of studying the metabolic stability, but this paragraph disrupts the logic connection of the abstract. 

8. P.2, “Upon reviewing the literature extensively, it was found that few articles were published for BOS”, this is not a good expression to display the significance of this work and should be deleted. 

9. P.3, “However, no method was developed for quantification of BOS in human liver microsomes (HLMs) have been reported” is not correct regarding its grammar. Please rephrase.

10. P.3, should be “in the presence of”.

11. P.3, should be “permits”.

12. P.3, should be “develop and validate”.

13. P.3, please improve wording on “Protein precipitation was applied for drug extraction from the HLM matrix”.

14. P.4, “The systemic name of BOS is 4-[(2,4-Dichloro-5-methoxyphenyl)amino]-6-methoxy-7-[3-(4-methyl-1-piperazinyl)propoxy]-3-quinolinecarbonitrile. Tofacetinib (TOF) was selected as IS in BOS quantification in the established LC-MS/MS analytical methodology. The IUPAC name of TOF is 3-{(3R,4R)-4-Methyl-3-[methyl(7H-pyrrolo[2,3-d]pyrimidin-4-yl)amino]-1-piperidinyl}-3-oxopropanenitrile”, what’s the meaning to show its chemical structure name?

15. P.4, does the text show “kwater”?

16. There are a lot of grammar & format mistakes in the following text. Please double check.

17. Is “water containing 0.1% formic acid and 30% acetonitrile (ACN)” the best composition? Have other ratios been optimized? 

Author Response

Authors’ response

We thank the editor this opportunity to improve our manuscript and be considered again for publication in Molecules Journal. We give below detailed answers to each question raised by reviewer # 2. We mark all reply to the comments by red color in the revised manuscript.

Reviewer # 2

Comments and Suggestions for Authors

This research article is aiming at developing a rapid LC-MS/MS method to characterize bosutinib after investigating calibration range, intra- and inter-day accuracy and precision and other analytical standards. Although the method design is straightforward and correct, this work lacks novelty, experimental details and relevance to this journal. In addition, this manuscript has various flaws in its English style and is not qualified as an acceptable form, especially some sentences are really redundant and confusing. Please see some detailed suggestions below.

Authors’ response

We appreciate the reviewer’s words and his/her suggestions to improve our manuscript. We give below our answer to his/her concerns.

Point # 1  

1-    P.1 title, “rapid validated” has a grammar mistake. This title is confusing with the combination of these three parts. Please rephrase.

Authors’ response

We appreciate the reviewer’s comment and his/her suggestions to improve our manuscript.

The manuscript has been edited to ensure language and grammar accuracy by Editage (a brand of Cactus Communication). We attached a certificate of English editing.

We updated the title to the following:

“Rapid LC-MS/MS Bosutinib Quantification with Applications on metabolic stability estimation”

  1. P.1 short title, should be “Bosutinib’s metabolic stability” or “The metabolic stability of Bosutinib”.

Authors’ response

We removed the short title and we redesigned the whole manuscript.

We changed the layout as recommended by the journal guidelines.

We removed the short title as it is not required by the journal.

  1. P.2, what’s meaning of putting its full name in the beginning of the paragraph while lacking any elaboration? “Bosutinib (BOS; bosutinib hydrate, 4‐[(2, 4‐Dichloro‐5‐methoxyphenyl) amino]‐6‐methoxy‐7‐[3‐(4‐methylpiperazin‐1‐yl) propyloxy] quinoline‐3‐carbonitrile monohydrate).”

 Authors’ response

We appreciate the reviewer’s comment and his/her suggestions to improve our manuscript.

We removed the full name of Bosutinib as requested as it has no scientific value in the current manuscript.

  1. P.2, the main text should also contain “Fig.1” which can give readers its reference.

Authors’ response

We cited figure 1 in the main text.

 “Bosutinib (BOS; Figure 1) is a second-generation dual Abl/Src inhibitor that shows potent inhibition for the growth of CML cells in vitro, is also active against multiple imatinib-resistant BCR-Abl mutations, and has proved efficacy in current clinical trials for imatinib-resistant CML”

  1. P.2, delete one “in” in “used approach in in vitro”.

Authors’ response

One in was removed as requested. Sorry for the typo mistake.  

  1. P.2, “the derived parameters can be used for extrapolation to various in vivo physiological parameters” sounds cryptic, please improve wording.

Authors’ response

We appreciate the reviewer’s comment and his/her suggestions to improve our manuscript.

The sentence was rephrased as requested:

 “The two parameters are computed following the “in vitro half-life” approach based on the “well-stirred” model [15,16]. As the “well-stirred” model is the most frequently used approach in the in vitro drug metabolism prediction, the derived in vitro calculated parameters can be used for the prediction of various in vivo physiological parameters, involving the potential toxicity and accumulation [17,18]”

  1. P.2, please rephrase and rearrange this paragraph in the main text, “Diarrhea is the most common side effect in patients taking bosutinib and concurrent antidiarrheal medication may reduce this side effect”. I understand that authors intend to show the importance of studying the metabolic stability, but this paragraph disrupts the logic connection of the abstract. 

Authors’ response

We appreciate the reviewer’s comment and his/her suggestions to improve our manuscript.

We rephrase the required sentences and we rearranged the whole paragraph and moved to other paragraph.

We updated the following paragraph in the revised manuscript:

“The Food and Drug Administration (19/12/2017), granted accelerated approval for bosutinib (BOSULIF) that was developed by Pfizer Inc. for the treatment of patients with newly diagnosed chronic phase (CP) Philadelphia chromosome-positive (Ph+) CML [9]. Bosutinib (BOS; Figure 1) is a second-generation dual Abl/Src inhibitor that shows potent inhibition for the growth of CML cells in vitro, is also active against multiple imatinib-resistant BCR-Abl mutations, and has proved efficacy in current clinical trials for imatinib-resistant CML [10–12]. The most common side effects of BOS is Diarrhea and can be avoided by concurrent antidiarrheal medication. Other minor side effects of BOS in-clude possible dermatological problems, grade 2 diarrhea, grade 1 fatigue associated with secondary dehydration caused by diarrhea, grade 1 skin rash, grade 1 AST elevation, and grade 2 vomiting that indicate the safety of BOS if compared to imatinib [13,14].”.

  1. P.2, “Upon reviewing the literature extensively, it was found that few articles were published for BOS”, this is not a good expression to display the significance of this work and should be deleted. 

Authors’ response

We appreciate the reviewer’s comment and his/her suggestions to improve our manuscript.

We deleted the required sentences and we rephrased the whole paragraph to display the significance of the work.

We updated the following paragraph in the revised manuscript:

“[20]. No analytical method has been developed for quantifying BOS in human liver microsomes (HLMs) or for the metabolic stability estimation of BOS. Accordingly, this study focuses on the quantification of BOS in spiked HLMs using tofacitinib as the internal standard (TOF; IS) over a very short run time (5 min), which permits its application in metabolic stability estimations. This study aimed to develop and validate a reliable LC-MS/MS method. Protein precipitation using acetonitrile (ACN) was used for analytes (BOS and TOF) extraction from the HLMs matrix.”.  

  1. P.3, “However, no method was developed for quantification of BOS in human liver microsomes (HLMs) have been reported” is not correct regarding its grammar. Please rephrase.

Authors’ response

We appreciate the reviewer’s comment and his/her suggestions to improve our manuscript.

The manuscript has been edited to ensure language and grammar accuracy by Editage (a brand of Cactus Communication). We attached a certificate of English editing.

We updated the sentence to the following:

“No analytical method has been developed for quantifying BOS in human liver microsomes (HLMs) or for the metabolic stability estimation of BOS. Accordingly, this study focuses on the quantification of BOS in spiked HLMs using tofacitinib as the internal standard (TOF; IS) over a very short run time (5 min), which permits its application in metabolic stability estimations. This study aimed to develop and validate a reliable LC-MS/MS method. Protein precipitation using acetonitrile (ACN) was used for analytes (BOS and TOF) extraction from the HLMs matrix.”

  1. P.3, should be “in the presence of”.

Authors’ response

The manuscript has been edited to ensure language and grammar accuracy by Editage (a brand of Cactus Communication). We attached a certificate of English editing.

We updated the sentence to the following:

“Accordingly, this study focuses on the quantification of BOS in spiked HLMs using tofacitinib as the internal standard (TOF; IS) over a very short run time (5 min), which permits its application in metabolic stability estimations. This study aimed to develop and validate a reliable LC-MS/MS method. Protein precipitation using acetonitrile (ACN) was used for analytes (BOS and TOF) extraction from the HLMs matrix.”.

  1. P.3, should be “permits”.

Authors’ response

The manuscript has been edited to ensure language and grammar accuracy by Editage (a brand of Cactus Communication). We attached a certificate of English editing.

We updated the sentence to the following:

“which permits its application in metabolic stability estimations”

  1. P.3, should be “develop and validate”.

Authors’ response

We updated the sentence to the following:

“This study aimed to develop and validate a reliable LC-MS/MS method”.  

  1. P.3, please improve wording on “Protein precipitation was applied for drug extraction from the HLM matrix”.

Authors’ response

We appreciate the reviewer’s comment and his/her suggestions to improve our manuscript.

“Protein precipitation using acetonitrile (ACN) was used for analytes (BOS and TOF) extraction from the HLMs matrix.”

  1. P.4, “The systemic name of BOS is 4-[(2,4-Dichloro-5-methoxyphenyl)amino]-6-methoxy-7-[3-(4-methyl-1-piperazinyl)propoxy]-3-quinolinecarbonitrile. Tofacetinib (TOF) was selected as IS in BOS quantification in the established LC-MS/MS analytical methodology. The IUPAC name of TOF is 3-{(3R,4R)-4-Methyl-3-[methyl(7H-pyrrolo[2,3-d]pyrimidin-4-yl)amino]-1-piperidinyl}-3-oxopropanenitrile”, what’s the meaning to show its chemical structure name?

Authors’ response

We appreciate the reviewer’s comment and his/her suggestions to improve our manuscript.

We removed the following two sentences:

TOF was selected as the IS for BOS quantification using the established LC-MS/MS analytical methodology. The IUPAC name of TOF is 3-{(3R,4R)-4-Methyl-3-[methyl(7H-pyrrolo[2,3-d]pyrimidin-4-yl)amino]-1-piperidinyl}-3-oxopropanenitrile.

We updated the paragraph to the following:  

“The in silico stability of BOS towards metabolism was determined utilizing the P450 metabolism module of the StarDrop software package from Optibrium Ltd. (Cambridge, MA, USA). To determine the metabolic lability of BOS, the site labilities of individual atoms can be combined to calculate composite site lability (CSL), which indicates the overall metabolic stability of BOS, as explained by equation (5):”

  1. P.4, does the text show “kwater”?

Authors’ response

We corrected the typo mistake: as kw is the rate constant for water formation

We updated the following paragraph in the revised manuscript:

“To determine the metabolic lability of BOS, the site labilities of individual atoms can be combined to calculate composite site lability (CSL), which indicates the overall metabolic stability of BOS, as explained by equation (5):

                                  (5)

where kw is the rate constant for water formation.

”.

  1. There are a lot of grammar & format mistakes in the following text. Please double check.

Authors’ response

The manuscript has been edited to ensure language and grammar accuracy by Editage (a brand of Cactus Communication). We attached a certificate of English editing.

  1. Is “water containing 0.1% formic acid and 30% acetonitrile (ACN)” the best composition? Have other ratios been optimized? 

Authors’ response

We updated the following paragraph in the revised manuscript:.

“LC analytical chromatographic parameters including the resolution of the target an-alytes (BOS and TOF), such as mobile phase composition, stationary phase nature, and pH were optimized. The mobile phase consisted of 30% ACN and 70% aqueous solution at a flow rate of 0.15 mL/min. Increasing ACN % generated overlapped peaks and a poor resolution, while decreasing ACN % generated long elution time. The pH of the 0.1% for-mic acid in water (aqueous solution) was 3.2 as an increased pH value caused long elu-tion time and a chromatographic peak tailing. Different stationary phases were examined, such as polar columns (HILIC columns). But, neither BOS nor TOF was retained on chro-matographic column, and the best results were achieved through the use of a C18 column (2.1×50 mm, 1.8 μm) (Agilent, USA) at 22±2 0C. Injection volume and run time were 5 min and 5 µL, respectively.”.

Thank you for your kind cooperation.

Kind regards

Mohamed W. Attwa, 

Dept. Pharm. Chem.,

College of Pharmacy,

King Saud University,

P.O. Box 2457,

Riyadh, 11451,

Saudi Arabia

Round 2

Reviewer 2 Report

Thank you for your time and this revised version looks better.